# *KRAS* Mutation Status in Bulgarian Patients with Advanced and Metastatic Colorectal Cancer

**DOI:** 10.3390/ijms241612753

**Published:** 2023-08-13

**Authors:** Maria Radanova, Galya Mihaylova, George St. Stoyanov, Vyara Draganova, Aleksandar Zlatarov, Nikola Kolev, Eleonora Dimitrova, Nikolay Conev, Diana Ivanova

**Affiliations:** 1Department of Biochemistry, Molecular Medicine and Nutrigenomics, Medical University of Varna, 9000 Varna, Bulgaria; galya.mihaylova@mu-varna.bg (G.M.); divanova@mu-varna.bg (D.I.); 2Department of Clinical Pathology, Complex Oncology Center—Shumen, 9700 Shumen, Bulgaria; georgi.geesh@gmail.com; 3Department of Surgery Diseases, Medical University of Varna, 9000 Varna, Bulgaria; vyara.draganova@mu-varna.bg; 4Second Clinic of Surgery, UMHAT “St. Marina”, 9000 Varna, Bulgaria; 5Department of General and Operative Surgery, Medical University of Varna, 9000 Varna, Bulgaria; aleksandar.zlatarov@mu-varna.bg (A.Z.); nikola.kolev@mu-varna.bg (N.K.); 6First Clinic of Surgery, UMHAT “St. Marina”, 9000 Varna, Bulgaria; 7Department of Oncology, Medical University of Varna, 9000 Varna, Bulgaria; eleonora.dimitrova@mu-varna.bg (E.D.); nikolay.conev@mu-varna.bg (N.C.); 8Clinic of Medical Oncology, UMHAT “St. Marina”, 9000 Varna, Bulgaria

**Keywords:** *RAS*, *KRAS*, colorectal cancer, left-sided colon cancer, right-sided colon cancer

## Abstract

*RAS* somatic variants are predictors of resistance to anti-EGFR therapy for colorectal cancer (CRC) and affect the outcome of the disease. Our study aimed to evaluate the frequency of *RAS*, with a focus on *KRAS* variants, and their association with tumor location and some clinicopathological characteristics in Bulgarian CRC patients. We prospectively investigated 236 patients with advanced and metastatic CRC. Genomic DNA was extracted from FFPE tumor tissue samples, and commercially available kits were used to detect *RAS* gene somatic mutations via real-time PCR. A total of 115 (48.73%) patients tested positive for *RAS* mutations, with 106 (44.92%) testing positive for *KRAS* mutations. The most common mutation in exon 2 was c.35G>T p.Gly12Val (32.56%). We did not find a significant difference in *KRAS* mutation frequency according to tumor location. However, patients with a mutation in exon 4 of *KRAS* were 3.23 times more likely to have a tumor in the rectum than in other locations (95% CI: 1.19–8.72, *p* = 0.021). Studying the link between tumor location and *KRAS* mutations in exon 4 is crucial for better characterizing CRC patients. Further research with larger cohorts, especially in rectal cancer patients, could provide valuable insights for patient follow-up and treatment selection.

## 1. Introduction

Colorectal cancer (CRC) is one of the most aggressive malignancies of the digestive system, and the prolongation of overall survival (OS) without impairing the life quality is still a main aim. In countries such as Bulgaria, where there are no large-scale screening programs for CRC, the incidence of the disease has been increasing [1]. A high percentage of patients are diagnosed at advanced stages, with a high likelihood of metastasis and a poor prognosis. Some targeted treatments for metastatic CRC (mCRC) involve the use of monoclonal antibodies that inhibit the signaling pathway initiated by the binding of the epidermal growth factor to its receptor (EGFR). The response to this therapy is determined by the presence of somatic mutations in genes that are part of the EGFR signaling cascade, such as *RAS* and *BRAF*. In our country, when a patient is diagnosed with mCRC, it is common practice to test for *RAS* mutations (genetic testing for *BRAF V600E* is not regularly included) as part of the treatment planning process. The lack of *RAS* mutations can indicate that the patients may benefit from EGFR monoclonal antibody (anti-EGFR) therapy combined with chemotherapy as a first-line treatment.

*RAS* mutational status has a proven prognostic significance, and therefore testing for *KRAS* (in exon 2, 3, and 4) and *NRAS* (in exon 2, 3, and 4) mutations is mandatory along with DNA mismatch repair (MMR)/microsatellite instability (MSC) status testing and *BRAF* (exon 15) genetic testing, according to the current international guidelines. Studies have shown that the frequency of *RAS* mutations in mCRC patients varies depending on the community [2,3,4,5,6,7]. The frequency of *KRAS* gene mutations ranges from 27% to 56%, while the frequency of *NRAS* mutations ranges from 1% to 7% [8]. Several studies have found an association between demographic data, clinicopathological features, and *RAS* mutational status. For instance, *KRAS* gene mutations have been linked to factors such as the patient’s sex, age, and tumor histological type [7,9,10,11]. A lower frequency of *KRAS* mutations is found in left-sided colon tumors compared to right-sided colon tumors [12,13,14,15]. Tumor localization has gained importance with the finding that anti-EGFR therapy combined with chemotherapy is more effective in wild-type *KRAS*, *NRAS*, and *BRAF* patients with left-sided primary CRC tumors [16]. It is already known that right-sided tumors derive limited benefits from first-line anti-EGFR treatment, even with wild-type *RAS* status [17].

*KRAS* mutations are known to be associated with poor prognosis in colorectal cancer patients, due to their role in determining drug resistance and their prevalence in right-sided colon tumors. As a result, testing for *KRAS* mutational status and assessing certain clinicopathological characteristics is essential for optimizing the follow-up and treatment of CRC patients. With this in mind, our study aimed to evaluate the frequency of *RAS* mutations (including both *KRAS* and *NRAS*) and to investigate a possible association between these mutations and certain clinicopathological characteristics, as well as tumor localization, in a group of Bulgarian patients with mCRC.

## 2. Results

### 2.1. Clinicopathological Characteristics of Patients’ Cohort

This is a prospective one-centered study of CRC patients in the III and IV stage. The mean age of the 236 CRC patients at the time of testing was 63.92 ± 10.52, with a range of 30–90 years. Female patients were 98 (41.53%), and male patients were 138 (58.47%). The mean age of women and men was almost the same (63.87 ± 11.13 vs. 63.96 ± 10.09). Selected CRC patients were divided into three groups depending on primary tumor localization—70 (29.66%) with a tumor in the right-sided colon, 91 (42.13%)—in the left-sided colon, and 75 (34.72%)—in the rectal colon.

Tumors in the ascending colon, hepatic flexure, and proximal two-thirds of the transverse colon were defined as right-sided tumors, whereas these in the distal one-third of the transverse colon, splenic flexure, descending colon, and sigmoid colon were defined as left-sided tumors.

In our cohort, right-sided tumors were a more common event for women than for men (OR *=* 1.80, 95% CI: 0.95–3.42, *p =* 0.071), but rectal cancer was more frequent in men in our cohort (OR *=* 2.11, 95%CI: 1.18–3.77, *p =* 0.012). Histological subtypes of tumors were two in number (Table 1). Mucinous adenocarcinoma was more commonly found in right-sided tumors (OR *=* 3.32, 95%CI: 1.21–9.14, *p =* 0.002) compared to left-sided and rectal tumors. Well-differentiated tumors were 23 (9.75%) G1, moderately differentiated tumors were 172 (72.88%) G2, and poorly differentiated G3 were 41 (17.37%). The patients’ Eastern Cooperative Oncology Group (ECOG) performance status was assessed to be <2. Liver metastasis was found in 79.66% (188/236) of patients. The second most common metastatic site was the lung, with 27.97% (66/236) of patients having lung metastases. Peritoneum metastasis was found in 16.95% (40/236) of patients. Additionally, 20.76% (49/236) had both lung and liver metastases, 9.32% (22/236) had both liver and peritoneum metastases, and 5.08% (12/236) had both lung and peritoneum metastases. In 31.78% (75/236) of patients, metastases were found in locations other than the liver, lung, or peritoneum.

The clinicopathological characteristics of the selected CRC patients are presented in Table 1.

### 2.2. Distribution and Frequencies of RAS Mutation in CRC Patients

A total of 115/236 (48.73%) of the patients were positive for RAS mutations, and 121/236 (51.27%) had wild-type RAS mutation status (Table 1). In total, 106 (44.92%) of CRC patients were positive for KRAS mutations. Five patients had two KRAS mutations simultaneously. A total of 86 (36.41%) KRAS mutations were in exon 2, 5 (2.12%) KRAS mutations were in exon 3, and 20 (8.47%) KRAS mutations were in exon 4, as detected in our study. Among the KRAS mutations in exon 2, those in codon 12 were more prevalent than the ones in codon 13 (31.33% vs. 5.08%, Table 2).

The most commonly registered mutation in exon 2 for our cohort was c.35G>T p.Gly12Val (28/86, 32.56%), followed by c.35G>A p.Gly12Asp (18/86, 20.93%). On the other side, the variant c.34G>T p.Gly12Cys (n *=* 4, 4.65%) was one of two the least represented, and it was not established in patients with right-sided cancer. All mutations in codon 13 of exon 2 were of the type c.38G>A p.Gly13Asp (Table 3).

Only one patient was positive for the KRAS exon 3 mutation/s in codon 59 (Table 2). Possible mutations could be c.175G>A p.Ala59Thr, c.176C>A p.Ala59Glu, or c.176C>G p.Ala59Gly because of “Easy^®^ KRAS” kit’s inability to distinguish between them. For the same reason, we could not point out the current mutation/s in KRAS exon 3/codon 61 positive patients. Possible mutations in 61 codon could be c.181C>A p.Gln61Ly, c.182A>T p.Gln61Leu, c.182A>G p.Gln61Arg, c.183A>C p.Gln61His, or c.183A>T p.Gln61His.

There were no patients positive for KRAS mutations in codon 117 of exon 4 (Table 3). All patients positive for KRAS exon 4 (n *=* 20) had mutations in codon 146, which could be c.436G>A p.Ala146Thr, c.436G>C p.Ala146Pro, or c. 437C>T p.Ala146Val.

From all (n *=* 130) patients investigated for NRAS mutations, eight patients (6.15%) harbored mutations. We detected 10 NRAS mutations in these samples, and two patients simultaneously had two NRAS mutations. Three (2.31%) samples had NRAS mutations in exon 2 (codon 12 or 13), seven (5.39%) samples in exon 3 (codon 59 or 61), and none in exon 4 (codon 146 or 117). Information on the prevalence of NRAS mutations in our cohort is incomplete because only KRAS wild-type samples were tested for the presence of NRAS variants.

### 2.3. Association between the Presence of KRAS Mutations and the Localization of Tumors

The distributions of mutations in exons of the KRAS gene in patients with different tumor localization are presented in Figure 1. Mutations in exon 2 were predominant in all three different localizations—right-sided, left-sided, and rectum. The comparison between codon 12 KRAS and codon 13 KRAS mutations according to tumor localization did not reveal significant differences (*p =* 0.430, Table 4). The percentages in Table 4 represent the proportion of patients with codon KRAS mutations among all KRAS-mutated patients with the corresponding tumor localization.

Patients in the presence of a mutation in exon 4 of the KRAS gene had tumors in the rectum 3.23 times more often than in the left, transverse, and right colon (95% CI: 1.19–8.72, *p =* 0.021).

### 2.4. Association between KRAS Mutational Status and Clinicopathological Characteristics of CRC Patients

Analysis of the relation between the KRAS mutational status and clinicopathological characteristics in the patients revealed that the presence or absence of KRAS mutations was not related to the sex of patients (*p =* 0.507), patients’ age at the time of testing (*p =* 1.000), histological type of tumor (*p =* 0.128), and simultaneous distant and lymph node metastasis (*p =* 0.471), while wild-type KRAS patients had more rare distant metastasis in comparison with patients with KRAS mutations (0.35, 95% CI: 0.19–0.64, *p =* 0.006, Table 5). KRAS mutations were found more frequently in right-sided tumors, with a prevalence of 53.52%. The comparison only between right-sided and left-sided tumors did not reach a significant difference (*p =* 0.085).

## 3. Discussion

*RAS* mutation frequency has been shown to vary between different racial groups and ethnic subgroups. These variations may be attributed to genetic differences between heterogeneously related races, or disparities arising from the different selection of patients’ cohorts and the use of different methods for mutation analysis. In a large-scale study of *RAS* testing practices in mCRC patients across Europe, the overall prevalence of *RAS* mutations was found to be 46.0% (with a range between 40.00% and 52.10%). Bulgaria was not among the 26 European countries included in this survey. In our study, we found a *RAS* mutation frequency of 48.73%, which is comparable to data from other studies of Caucasians, tested using real-time PCR.

KRAS mutations have been found to occur at different frequencies in different racial groups. They are least common in Asians, followed by Caucasians, and are most common in African Americans. Since colorectal cancers with *KRAS* mutations are a heterogeneous group, the frequency of these mutations can also vary among different populations. In our study, the prevalence rate of *KRAS* mutation was 44.93%, which is similar to data reported for populations in Slovenia (48.8%) [18], Germany (41%) [19], Turkey (44%, 41.9%, and 47.6%) [20,21,22], Italy (43%, 47%) [23,24,25], Russia (49.5%) [26], and Romania (45.2%) [27]. Some of these countries are geographically close to our location.

In various studies, including ours, the highest rates of *KRAS* mutation prevalence were found for exon 2, codon 12, and codon 13 mutations. The most frequent mutation among our patients was c.35G>T p.Gly12Val (32.56%), followed by c.35G>A p.Gly12Asp (20.93%). These two *KRAS* mutations in codon 12, as well as the codon 13 c.38G>A (p.Gly13Asp) mutation, are the most common mutations among the European population as G12D and G12V are vying for the top spot [28,29,30,31]. The poor prognosis associated with exon 2 *KRAS* mutations is well documented, but the prognostic or predictive value of mutations in codons 12 and 13 remains a subject of debate. While some studies have reported that codon 12 mutations are associated with a worse prognosis [32,33,34], others found that patients with codon 13 mutations have a significantly worse prognosis compared to those with either wild-type *KRAS* or codon 12 mutations [35,36]. However, many studies have found no significant difference in prognosis between patients with codon 12 and codon 13 mutations [37,38]. The varying outcomes of these studies could be attributed to factors such as the size of the cohort, the methods used for data analysis, or the race of the participants. Alternatively, the discrepancies may stem from the fact that codon 12 and codon 13 *KRAS* mutations have distinct biochemical properties and exhibit tissue- and treatment-specific mutational patterns [39,40]. Patients with exon 2 *KRAS* mutations face limited options for targeted treatment due to resistance to anti-EGFR therapy. However, the success of *KRAS* G12C inhibitors in treating non-small cell lung cancer (NSCLC) patients offers hope for similar results in CRC patients. G12C KRAS mutations occur in 3–4% of CRC patients, including four patients with G12C *KRAS* mutated mCRC in our cohort. These patients have worse overall survival than patients without G12C *KRAS* mutations, and there are currently no approved treatments specifically targeting the G12C *KRAS* for CRC [41]. However, a recent study reported effective therapy with adagrasib, an oral RAS GTPase inhibitor, alone and in combination with cetuximab in heavily pretreated patients [42]. Since adagrasib showed greater biological activity when combined with cetuximab, combination therapy may be the key to improved response.

*KRAS* mutations in exons 3 and 4 have not been as extensively studied as those in exon 2 due to their low mutation rate, leaving their clinical relevance unclear. Additionally, patients with *KRAS* mutations in exons 3 or 4 are often analyzed as a single group due to the small sample size. However, Guo et al. (2021) and Levacchi et al. (2022) recently reported significant differences in prognosis between *KRAS* exon 3 and exon 4 mutations, with exon 4 mutations predicting the best prognosis and exon 3 mutations the worst [35,43]. In our patient cohort, we found that *KRAS* exon 4 mutations were significantly associated with rectal tumor localization. Several studies have reported that rectal cancer has a better prognosis than colon cancer [31,44,45]. Thus, it is possible that the location of the tumor, rather than the type of codon-specific *KRAS* mutation, plays a more significant role in this case.

In our study, we were unable to evaluate patient outcomes or compare the clinical course of the disease between patients harboring different *KRAS* mutations. We only confirmed the well-established relationship between wild-type *KRAS* status and a less aggressive course of CRC. Our most significant finding was the higher prevalence of exon 4 *KRAS* mutations in rectal tumors compared to tumors in other colon segments (OR = 3.23, 95% CI: 1.19–8.72, *p* = 0.021).

The prevalence of *RAS* and *KRAS* mutations in the individual anatomical locations of the tumors in CRC patients is the other controversial issue on the subject. While some studies have reported that *KRAS* mutations are more common in right-sided colon tumors [13,46], others have found them to be more frequent in left-sided tumors [47,48,49]. Still, others have not found a significant difference in mutation frequency according to tumor localization [7,22,50]. Our results place our study in the third group. However, the literature predominantly reports a higher frequency of *KRAS* mutations in right-sided tumors, a trend we also reported.

Accurately determining the tumor’s localization and *KRAS* mutational status is crucial for selecting the appropriate therapy and predicting the response to treatment. There are two major challenges in addressing this issue. The first is accurately defining the boundaries between distinct anatomical regions in the colon and establishing consistent definitions for left- and right-sided tumors. This is because several studies have emphasized the differences in epidemiological and clinicopathological characteristics between right-sided and left-sided CRC, leading to their classification as separate diseases, with the emphasis that right-sided tumors have a poorer prognosis than their left-sided counterparts [13,51]. The second challenge is the variable nature of *RAS* mutations. Tumors with wild-type *KRAS* status at diagnosis may develop *KRAS* mutations during treatment, potentially due to acquired resistance to EGFR-directed therapies or due to the development of mutations in the EGFR extracellular domain that prevent therapeutic agent binding [25]. In this regard, it is interesting to note that the use of antiangiogenic agents in combination with chemotherapy as a first-line treatment may revert *RAS*-mutated tumors to wild type, allowing patients to receive EGFR inhibitors as a second-line therapy [52,53,54]. Liquid biopsy, which is entering recently for screening for *RAS* mutations, provides a means to monitor the dynamics of the mutational status during the course of therapy [55].

In addition to comparing left-sided and right-sided colon cancers, a comparison is also made between colon cancers, particularly left-sided and rectal cancers. Most studies have found that the frequency of exon 2 *KRAS* mutations in the rectum is similar to that of left-sided colon cancers and lower than that of right-sided colon cancers [25]. In our study, the prevalence of exon 2 *KRAS* mutations was lower in the rectum (23/36, 63.88%) compared to the right-sided colon (32/38, 84.21%), and also lower than in the left-sided colon (31/37, 83.78%). However, these differences were not statistically significant. The impact of *KRAS* mutations on rectal cancer is not yet fully understood, and there is limited data on mutations in exons 3 and 4 due to their low occurrence rate.

Our pilot study is the first to assess the frequency of *RAS* mutations in the Bulgarian population. However, its findings need to be confirmed with a larger sample size for a more accurate representation. Another limitation of our study was the use of a PCR-based method, which while sensitive, did not allow for discrimination between different variants in exon 3 (codons 59 and 61) and particularly in exon 4 (codons 117 and 146) of the *KRAS* gene. We plan to overcome the second limitation by using next-generation sequencing (NGS) for validating the analysis in our future analyses. Although NGS is currently time-consuming and expensive, it can report all mutations present in the analyzed exons of the *RAS* genes and test other genes related to therapy or prognosis of CRC such as in *PIK3CA* or in *TP53* genes. The introduction of liquid biopsies for disease monitoring also allows for the routine use of digital PCR, which is a suitable choice when there is low tumor cellularity [56]. Additionally, it would be interesting to investigate any potential associations between *KRAS* mutational status and other clinicopathological characteristics not collected for our cohort, as well as survival and recurrence rates.

## 4. Materials and Methods

### 4.1. Patients

A total of 236 Bulgarian patients with metastatic colorectal cancer (mCRC) were investigated prospectively. The patients were selected in University Hospital “St. Marina”, Varna, from September 2017 to August 2020. The including criteria of the study were as follows: the diagnosis of patients—mCRC; the availability of complete clinical information of the patients; the presence of a histological material from surgery; and a signed informed consent form for genetic analysis and the processing of personal data from patients. The study was prospective and had the approval of the Ethics Review Boards of the Medical University of Varna (protocol №66/18 August 2017). Each selected patient signed a consent form upon enrolment.

### 4.2. DNA Extraction

Genomic DNA was extracted from 5 μm thick sections of formalin-fixed paraffin-embedded (FFPE) tumor tissue samples using a QIAamp FFPE Tissue Kit (Qiagen, Hilden, Germany) according to the manufacturer protocol. Each section had enough material for the isolation of at least 100 ng of genomic DNA. Prior to DNA isolation, the fixed samples were deparaffinized with xylene and washed in 95% ethanol to extract residual xylene. DNA was eluted in 60 μL ATE buffer and stored at −20 °C for further testing. A NanoDrop™2000 system (Thermo Fisher Scientific, Wilmington, DE, USA) was used for the evaluation of DNA concentration.

### 4.3. RAS Mutational Analysis

The commercially available CE-IVD “Easy^®^ KRAS” and “Easy^®^ NRAS” kits (Diatech Pharmacogenetics, Jesi (AN) Italy) were used for the detection of somatic mutations of the *KRAS* and *NRAS* genes by real-time PCR (Applied Biosystems™ 7500 Real-Time PCR Systems, Waltham, MA, USA). The two kits cover mutations in codons 12 and 13 (exon 2), codons 59 and 61 (exon 3), and codons 117 and 146 (exon 4) in both genes. The analysis performance required at least 20 ng of extracted genomic DNA. The amplification conditions were as follows: initial incubation step at 95 °C for 2 min, followed by 40 cycles at 95 °C for 10 s and at 58 °C for 60 s. The reaction volume was 25 μL.

### 4.4. Statistical Analysis

The data were analyzed using GraphPad Prism 6 software. Baseline characteristics in different groups were calculated using descriptive statistics. Fisher’s exact test and the chi-squared test were used to compare the presence of gene mutations with the clinicopathological characteristics of patients. Odds ratios (ORs) with 95% confidence intervals (CIs) for categorical outcomes were calculated using a binary logistic regression model. Two-tailed *p*-values (<0.05) were considered significant.

## 5. Conclusions

Understanding the relationship between tumor location and *KRAS* mutations in the less-studied exon 4 is essential for improving the characterization of CRC patients. Further investigation and confirmation of this association in a larger cohort, particularly among rectal cancer patients, would provide valuable insights for patient follow-up and treatment selection, given the significant differences in disease prognosis between patients with *KRAS* mutations in exons 3 and 4, as reported in the literature.

## Figures and Tables

**Figure 1 ijms-24-12753-f001:**
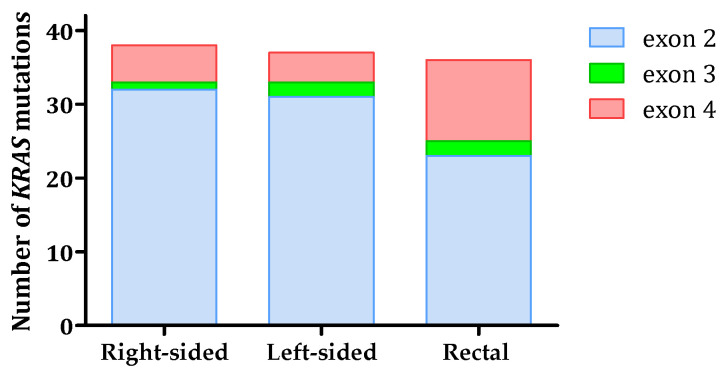
Mutations in exons 2, 3, and 4 of the *KRAS* gene in the different tumor localizations.

**Table 1 ijms-24-12753-t001:** Clinicopathological characteristics of CRC patients.

Clinicopathological Characteristics		Number (%)
Sex	Female	98 (41.53)
	Male	138 (58.47)
Age	<65	113 (47.88)
	≥65	123 (52.12)
Tumor localization	Right-sided	70 (29.66)
	Left-sided	91(38.56)
	Rectum	75 (31.78)
Histological type of tumor	Tubular adenocarcinoma	219 (92.80)
	Mucinous adenocarcinoma	17 (7.20)
*RAS* mutational status	Wild-type	121 (51.27)
	Mutated	115 (48.73)
TNM stage	III	72 (30.51)
	IV	164 (69.49)
Grade	G1	23 (9.75)
	G2	172 (72.88)
	G3	41 (17.37)

**Table 2 ijms-24-12753-t002:** Frequencies of *KRAS* mutations in CRC patients.

Exons	Number (%)	Codons	Number of Mutations (%)
Exon 2	86 (36.41)	Codon 12	74 (31.33)
		Codon 13	12 (5.08)
Exon 3	5 (2.12)	Codon 59	1 (0.42)
		Codon 61	4 (1.70)
Exon 4	20 (8.47)	Codon 117	1 (0.00)
		Codon 146	20 (8.47)

**Table 3 ijms-24-12753-t003:** Type of mutations in exon 2 of the *KRAS* gene detected in CRC patients.

Codons	Type of Mutations *	Number of Mutations (%)
Codon 12	c.34G>A p.Gly12Ser	10 (11.63)
	c.34G>C p.Gly12Arg	4 (4.65)
	c.34G>T p.Gly12Cys	4 (4.65)
	c.35G>A p.Gly12Asp	18 (20.93)
	c.35G>C p.Gly12Ala	10 (11.63)
	c.35G>T p.Gly12Val	28 (32.56)
Codon 13	c.38G>A p.Gly13Asp	12 (13.95)

* HGVS nomenclature, version 20.05 was used.

**Table 4 ijms-24-12753-t004:** Comparison of exon 2 *KRAS* mutations in codons 12 and 13 in different tumor localizations.

Tumor Localization in Patients with Exon 2 *KRAS* Mutations	Codon 12	Codon 13	*p*-Value
	Number (%)	Number (%)	
Right-sided (32/38)	28 (73.68)	4 (10.53)	0.430
Left-sided (31/37)	28 (75.68)	3 (8.11)	
Rectum (23/36)	18 (50.00)	5 (13.89)	

**Table 5 ijms-24-12753-t005:** Relationships between *KRAS* mutational status and clinicopathological characteristics of CRC patients.

Clinicopathological Characteristics		KRAS Mutational Status		*p*-Value
		Wild-Type, n (%)	Mutation, n (%)	
Sex	Female	51 (52.04)	47 (47.96)	0.507
	Male	79 (57.25)	59 (42.75)	
Age	<65	63 (54.78)	52 (45.23)	1.000
	≥65	67 (55.37)	54 (44.63)	
Tumor localization	Right-sided	33 (46.48)	38 (53.52)	0.214
	Left-sided	56 (60.22)	37 (39.78)	
	Rectum	41 (53.25)	36 (46.75)	
Histological type of tumor	Tubular adenocarcinoma	124 (56.62)	95 (43.38)	0.128
	Mucinous adenocarcinoma	6 (35.29)	11 (64.71)	
Distant metastasis	Yes	78 (47.56)	86 (52.44)	0.006
	No	52 (72.22)	20 (27.78)	
Distant and lymph node metastasis	Yes	108 (54.00)	92 (46.00)	0.471
	No	22 (61.11)	14 (38.89)	

## Data Availability

The data presented in this study are available on request from the corresponding author at e-mail: maria.radanova@gmail.com. The data are not publicly available due to their containing information that could compromise the privacy of research participants.

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
