# Peer review of "KRAS Mutation Status in Bulgarian Patients with Advanced and Metastatic Colorectal Cancer"

_ijms, 2023, doi:10.3390/ijms241612753_

Round 1

Reviewer 1 Report

In this manuscript, Maria Radanova et al; investigating the frequency of KRAS mutation in Hungarian population for advanced and metastatic colorectal carcinomas (mCRC). By using PCR based method, these authors determine the frequency of KRAS mutations in exon 2, 3 and 4 of 236 patients with mCRC. Mutations of KRAS were found in 106 cases and 9 others found to have NRAS mutations. The most common mutations found on exon 2. No correlation between tumor location and type of mutations were found, but patients with exon 4 have more chance of having rectum tumors than others. investigating the relationship between tumor location and KRAS mutations may provide tools for improving the characterization and treatment of CRC patients. Overall, the study is not novel, but provided clinical data to be used for classification of tumors that could impact the outcome of therapy.

Major criticism;

The results provided indeed important for defining the right treatment, but not very novel. Data nearly similar to statistics for other countries specifically the collective European union. It would be important to include additional data that cover other characteristics such as survival, mutation of other CRC genes such as p53.

While the authors used PCR based methods to identify mutations within exons, it would be important to discuss other alternative methods to complement the data.

May need minor english edditing.

Author Response

Reviewer 1

Comment:

In this manuscript, Maria Radanova et al; investigating the frequency of KRAS mutation in Hungarian population for advanced and metastatic colorectal carcinomas (mCRC). By using PCR based method, these authors determine the frequency of KRAS mutations in exon 2, 3 and 4 of 236 patients with mCRC. Mutations of KRAS were found in 106 cases and 9 others found to have NRAS mutations. The most common mutations found on exon 2. No correlation between tumor location and type of mutations were found, but patients with exon 4 have more chance of having rectum tumors than others. investigating the relationship between tumor location and KRAS mutations may provide tools for improving the characterization and treatment of CRC patients. Overall, the study is not novel, but provided clinical data to be used for classification of tumors that could impact the outcome of therapy.

 Major criticism;

The results provided indeed important for defining the right treatment, but not very novel. Data nearly similar to statistics for other countries specifically the collective European union. It would be important to include additional data that cover other characteristics such as survival, mutation of other CRC genes such as p53.

Author’s Reply:

We would like to thank you for your thoughtful comments and constructive suggestions, which helped to improve the quality of this manuscript.

The aim of our study was to provide new data that are missing for our country, Bulgaria. We are not included in large-scale studies of European countries because there is no consolidated data on RAS mutations in our country. Our study offers an opportunity to create such a data set.

We appreciate your suggestion to include survival data. Unfortunately, we cannot perform the log-rank test for survival analysis because such data are not available in our database. We do not have access to official information on the patients’ survival. We do not have an ethical permit to request the de novo acquisition of such data. Moreover, most of these patients were surgically treated in our hospital, but are followed up in other medical centers. Therefore, most of the patients in our cohort are lost to follow-up. We acknowledge the lack of survival data as a limitation of our study in the manuscript.

A challenge for our laboratory is to introduce routine testing of BRAF and PIK3CA, along with RAS testing, because of the known success of the triple combination therapy with BRAF inhibitors, anti-EGFR monoclonal antibodies and PI3K inhibitors as a rational treatment regimen for metastatic colorectal cancer patients with KRAS/NRAS/BRAF wild type/PIK3CA mutated disease.

Unfortunately, the analysis of gain-of-function mutations in TP53 in patients in our hospital could only be a subject of our future scientific interest, because, although they are prognostically informative, they have not been effectively targeted for therapeutic purposes.

While the authors used PCR based methods to identify mutations within exons, it would be important to discuss other alternative methods to complement the data.

Author’s Reply:

Thank you for the valuable suggestion to enrich the text with a discussion of alternative methods for identifying RAS mutations. The method we use to detect mutations is part of the PCR-based methods group. Although Easy®KRAS/NRAS is CE-IVD, it has limitations, such as the inability to distinguish mutations in exon 3 and in exon 4 of KRAS gene. We plan to overcome these limitations by introducing next-generation sequencing (NGS) for validating the analysis in a larger cohort. Although NGS is currently time-consuming and expensive, it can report all mutations present in the analyzed exons of the RAS genes and test other genes related to the disease. The inclusion of NGS in our future work will allow us to implement your other recommendation to study mutations in TP53.

In section "Discussion" in agreement with your recommendation we added a paragraph to comment other alternative methods to complement the data (lines from 275 to 284).

We are very thankful for your observations and comments. We are hoping that we have understood your comments and our answers are acceptable.

Reviewer 2 Report

In this study, the authors reported the percentage of KRAS mutations among CRC patients in Bulgaria. Then the author focused on the exact presence of these mutations , as reported mostly in codon 12 and 13. The authors then correlated between these mutations and patients demographic and some clinical data

Major points

1) Medical history of patients is missing and should be reported and correlated with the current findings.

2) As in table 5: only distant metastasis is significant when compared WT and mutation. The authors need to give details about metastasis sites.

3) Age: why the authors select 65 as the main age and distributed the patients according to age (below 65 and > 65).

4) Colon sections and IHC/ IF data about KRAS mutations are missing.

5) Extensive language editing is required

Minor

In the abstract: RT-PCR is the abbreviation of reverse transcription PCR, not real time PCR.

Extensive language language editing

Author Response

Reviewer 2

Comment:

In this study, the authors reported the percentage of KRAS mutations among CRC patients in Bulgaria. Then the author focused on the exact presence of these mutations , as reported mostly in codon 12 and 13. The authors then correlated between these mutations and patients demographic and some clinical data

Major points

1) Medical history of patients is missing and should be reported and correlated with the current findings.

Author’s Reply:

We would like to thank you for your thoughtful comments and constructive suggestions, which helped to improve the quality of this manuscript.

We agree with you that including more clinical data on patients would be useful for the study. However, due to the initially staked design of the study, we do not have an ethical permit to request the de novo acquisition of additional data from the patients’ medical history. All clinical data that were provided for our patients during the study are presented and analyzed in relation to the KRAS mutations in the manuscript. We concede the lack of other clinicopathological characteristics as a limitation of our study in the manuscript.

2) As in table 5: only distant metastasis is significant when compared WT and mutation. The authors need to give details about metastasis sites.

Author’s Reply:

Since some patients have more than one metastatic site, we did not perform a detailed analysis of the association of each type of distant metastatic site with KRAS mutations. The number of patients in the individual groups is small and this compromises the statistics. You may have noticed that we do not discuss the association of distant metastasis and mutational status in the manuscript, even though it is significant, because we cannot provide detailed information. However, following your recommendation, we have included details about metastatic sites in our patients in section “Results”, 2.1. Clinicopathological characteristics of patients’ cohort (lines from 104 to 110).

3) Age: why the authors select 65 as the main age and distributed the patients according to age (below 65 and > 65).

Author’s Reply:

Age is a factor that affects the risk of colorectal cancer. People who are older than 50 years have a higher chance of developing colorectal cancer. The average age when colon cancer is diagnosed is 68 for men and 72 for women. The average age when rectal cancer is diagnosed is 63 for both men and women. Many studies use 65 years as a cut-off point to define older adults. We also followed this definition to divide the patients by age because we aimed to compare our results with other study findings.

Reference

https://www.cancer.net/cancer-types/colorectal-cancer/risk-factors-and-prevention

4) Colon sections and IHC/ IF data about KRAS mutations are missing.

Author’s Reply:

Thank you for raising this question. As seen in the presented manuscript (exon mutations), KRAS mutation analysis was performed via molecular analysis on representative tumor tissue from the paraffin-embedded tissue section from the initial histopathological diagnosis, with representative being defined as at least 1mm2 of tissue with at least 50% of the cells within the specimen representative of the tumor, with its overall dominating histopathological type. As such, no IHC or IF test was performed to evaluate the mutational status as they have significantly more room for error, both in their preanalytical and analytical stages, when compared to direct molecular testing. Therefore, such kinds of testing are not applied at our hospital regarding RAS status.

5) Extensive language editing is required

Author’s Reply:

Thank you for your comment. We improved the quality of the English language as much as we could and we believe that we now meet the high standards of the journal.

Minor

In the abstract: RT-PCR is the abbreviation of reverse transcription PCR, not real time PCR.

Author’s Reply:

A technical error was made which has been corrected.

We are very thankful for your observations and comments. We are hoping that we have understood your comments and our answers are acceptable.

Reviewer 3 Report

The authors aim to study the prevalence of KRAS mutation status in Bulgarian patients with advanced colorectal cancer ( stage III and IV). The only valuable finding is that the KRAS mutation prevalence was similar to other European countries, with no relation to age, sex, location of tumor , etc.

The main limitation , also aknowledged by the authors is that they do not correlate Kras mutations with survival , response to treatment or other clinico-pathological features. 

Another question I have is why only patients with advanced colorectal cancer were selected?

The Discussions could be enriched with more recent studies on this topic.

Author Response

Reviewer 3

Comment:

The authors aim to study the prevalence of KRAS mutation status in Bulgarian patients with advanced colorectal cancer ( stage III and IV). The only valuable finding is that the KRAS mutation prevalence was similar to other European countries, with no relation to age, sex, location of tumor , etc.

The main limitation , also aknowledged by the authors is that they do not correlate Kras mutations with survival , response to treatment or other clinico-pathological features. 

Author’s Reply:

The main aim of our study is to report data about the frequency of KRAS mutations in a Bulgarian cohort, because no such study exists. The clinical information we had on the patients allowed us to analyze the presence of the mutations only with certain clinicopathological characteristics. Due to the initially staked design of the study, we do not have an ethical permit to request the de novo acquisition of additional data from the patients’ medical history. We also noted that our study is a pilot study, which opens perspectives for investigating the frequency of other mutations related to therapy or prognosis of CRC such as in PIK3CA or in TP53 genes; as well as conducting the log-rank test to assess differences in survival between patients harboring mutations and wild-type patients in our future work.

Another question I have is why only patients with advanced colorectal cancer were selected?

Author’s Reply:

In Bulgaria, RAS testing is mainly performed on patients with locally advanced cancer, inoperable and those in stage IV of the disease. This relates to the accepted standards of treatment for these patients. Therefore, the patients studied by us are in these two categories.

The Discussions could be enriched with more recent studies on this topic.

Author’s Reply:

Thank you for your suggestion. We have added five new sources, including a forthcoming article by authors who study the RAS mutations frequency in a neighboring country of ours.

We are very thankful for your observations and comments. We are hoping that we have understood your comments and our answers are acceptable.

Round 2

Reviewer 1 Report

These authors addressed my comments and this paper now acceptable for publication.

Reviewer 2 Report

No further comments

Moderate language editing

Reviewer 3 Report

The authors responded to all queries and improved their discussion section.